# Community perspectives of heat and weather warnings for pregnant and postpartum women in Kilifi, Kenya

Adelaide Lusambili[1,2]*, Veronique Filippi[3], Britt Nakstad[4,5], Julian Natukunda[6], Cathryn E. Birch[7], John H. Marsham[7], Nathalie Roos[8], Peter Khaemba[2], Sari Kovats[6]

1 Environmental Health And Governance Centre, School of Business, African International University, Nairobi, Kenya, 2 Institute for Human Development, Aga Khan University, Nairobi, Kenya, 3 Faculty of Epidemiology and Population Health, London School of Hygiene and Tropical Medicine, London, United Kingdom, 4 Department of Pediatric and Adolescent Health, University of Botswana, Gaborone, Botswana, 5 Forwerly; Division of Paediatric and Adolescent Medicine, Institute of Clinical Medicine, University of Oslo, Oslo, Norway, 6 Department of Public Health, Environments and Society, London School of Hygiene & Tropical Medicine, London, United Kingdom, 7 School of Earth and Environment, University of Leeds, Leeds, United Kingdom, 8 Department of Medicine, Clinical Epidemiology Division, Karolinska Institutet, Stockholm, Sweden

* adelaidelusambili@gmail.com

**Data Availability Statement:** Our study is qualitative, involving a small group of participants from rural areas. Due to the sensitive nature of the data and our commitment to maintaining

## Abstract

### Background

Extreme weather is a recognised risk factor for stillbirth and preterm birth, disrupts women's access to healthcare during pregnancy and childbirth, and negatively affects the care of newborns. Reliable and accessible heat and weather warning systems are key in alerting individuals to undertake protective measures. There is a notable gap in understanding how women and caregivers in rural East Africa perceive and utilize weather information. We investigated community members' heat and weather warning information-seeking behaviour, identified available sources, assessed their reliability and utility, and examined their influence on behaviour.

### Settings

Our research was conducted in rural Kilifi County in Kenya's coastal region. The area experiences temperatures exceeding 23°C throughout the year, with extended periods of extreme temperatures [> 40°C] and long and severe droughts.

### Methods

We conducted in-depth interviews [IDI] with pregnant and postpartum women [n = 21] and held six focus group discussions [FGDs] involving mothers-in-law and community health volunteers [CHVs]. The data were analysed in NVivo 12 using both inductive and deductive approaches.

confidentiality, releasing the data could potentially identify individuals. As a result, data access will only be available upon reasonable request through the Aga Khan University Research Ethics Office. For inquiries, please contact: Aga Khan University Research Ethics Office Email: research.office@aku.edu.

**Funding:** This work was supported by the Natural Environment Research Council (NERC) [grant numbers NE/T013613/1, NE/T01363X/1]; Research Council of Norway (RCN) [grant number 312601]; The Swedish Research Council for Health, Working Life and Welfare in collaboration with the Swedish Research Council (Forte) [grant number 2019-01570]; and the National Science Foundation (NSF) [grant number ICER-2028598]; coordinated through a Belmont Forum partnership. The funders had no role in study design, data collection and analysis, decision to publish, or manuscript preparation.

**Competing interests:** The authors declare no competing interests.

## Results

We found significant gaps concerning pregnant and post-partum women, and their caregivers, having timely access to weather forecasts and heat information from health or meteorological authorities. Information on heat and weather warnings is disseminated through various channels, including television, radio, mobile phones, and word of mouth, which are facilitated by community influencers such as CHVs and local chiefs. Indigenous methods of weather forecasting, such as cloud observation, consulting local "rainmakers", and studying the behavioural patterns of amphibians, are employed in conjunction with warnings from the Kenyan Meteorological Department (KMD). Barriers to accessing weather information include the cost of television and smartphones and a lack of segmented information in local languages.

## Conclusions

National and county meteorological services need to enhance public participation, communication, and the delivery of heat and weather information to guide community-level response measures and individual behaviour change. They should also collaborate with health professionals to address heat risks for vulnerable groups. Further research is needed to empower indigenous weather predictors with modern weather information and revise national policies to deliver tailored messages to vulnerable populations like pregnant and postpartum women.

## Introduction

It's widely acknowledged that Africa has limited access to weather warning systems [1]. This has motivated the United Nations Early-Warning-For-All initiative, which aims to provide universal access to disaster early warning systems by 2027. The initiative focuses on using technologies to build national capacity to increase coverage of warnings, and improve timely, accurate alerts and preparedness [2–4]. Weather warnings could provide cost-effective and reliable mechanisms for protecting lives and livelihoods from heatwaves, storms and other extreme weathers [5, 6]. Heat waves are under-reported in Africa [7–9], despite the strong evidence that heat adversely affects the health of pregnant women, increases the risk of poor birth outcomes, and is harmful to children, especially very young infants [10–14]. Many women in sub-Saharan Africa [SSA] are exposed to high environmental temperatures due to the high prevalence of traditional livelihood practices such as farming, herding, water well drawing, firewood fetching, and trading, all outdoor practices [11, 12, 15]. Heat stress exposures are projected to increase with climate change [1, 16]. Heat early warning systems can reduce avoidable human health consequences by alerting vulnerable individuals or their caregivers to take preventative measures and increasing awareness of heat risks [17–19]. These are essential parts of adaptation to climate change to protect human health [20].

Weather warning systems can be categorised into modern early warning systems and local/indigenous knowledge systems. A range of forecasts and weather warning systems have been developed by National Weather Services [NWS] in many countries, including weather forecasts, severe weather warnings, biometeorological warnings, sub-seasonal and seasonal forecasts. However, in Africa, weather and general climate information is often not available and

where it does exist, it is inaccessible to those that need it most [3, 21, 22]. Africa has lower implementation rates of early weather warning systems than other parts of the world [1, 3]. In most cases, weather warning information systems for heat or hot weather address drought conditions and target farmers, forestry and water source management, but the warnings are not related to health in general or health systems [23–25]. Significant gaps exist in the provision of location-specific, timely, and user-friendly climate and daily seasonal weather forecast information that effectively addresses the needs of local people at village levels [20, 26].

In poorly-resourced settings, people at lower socioeconomic levels rely on local or indigenous knowledge to anticipate threatening and adverse weather changes. Local knowledge indicators include the behaviour of animals, birds, insects, and celestial bodies such as the sun, moon, and stars [27–29]. Research shows that weather warning systems using local indigenous knowledge provide location-specific information, are easy to access, are trusted, and consistently produce knowledge [21, 30, 31]. However, these systems cannot be evaluated due to the lack of records of past weather warnings [23, 32].

The Kenyan Meteorological Department [33] issues daily temperature forecasts and a biometeorological forecast for malaria outbreaks; however, there is currently no operational heat health weather warning system. In this paper, we report findings from our study in Kilifi, which aimed to investigate community members' early weather information-seeking behaviour, identify available sources, assess their reliability and utility, examine their influence on behaviour, and explore measures available for supporting pregnant women in post-extreme weather.

## Methods

### Study site

We conducted our research in the coastal rural-sub counties of Kaloleni and Rabai of Kilifi County. The area is remote, with inadequate transportation and health facility infrastructure. Communities and health facilities are sparsely distributed, meaning many residents walk more than four kilometres to access health services [34, 35]. Average daily temperatures range between 23˚C and 39˚C, with the hottest period of the year between February and April. Around 70% of the population in the county lives below the poverty line compared to the national rate of 47% [34–36]. Households are patrilineal, polygamy is widely practised, and many girls are married before the age of 18 [37]. Communities practice Christianity, Islam or traditionalism and co-exist well with few conflicts. Our research population mainly spoke the Swahili and Giriama languages. More information about the Kilifi settings has been reported in the following papers [10, 12, 34, 37, 38]. Researchers in this study observed that houses are small, with low roofs constructed from either local materials, such as "makuti" grass, or iron sheets. Many houses lack windows, resulting in extremely high indoor temperatures throughout the hot season, mainly when cooking activities are conducted indoors [13]. Our research was conducted during a two-year drought period characterized by substantial crop failures, significant livestock losses, and the depletion of water sources.

### Study design

Our study employed qualitative approaches using IDIs and FGDs. Qualified and trained social science researchers collected data between February and April 2021, the hottest months, using a structured interview guide allowing open questions.

## Study participants' recruitment, inclusion, and exclusion criteria

Study researchers recruited participants using community networks developed by Aga Khan University, which has worked in the area for over a decade. These established social networks meant the researchers had already collaborated with some CHVs, who were instrumental in helping researchers to access health facilities and geographically dispersed villages to recruit eligible participants. These CHVs leveraged their frequent home visits and fluency in the local Giriama language to facilitate the smooth recruitment of participants by explaining the study objectives.

In Kilifi and Kaloleni, villages are spread out, and each village has a local health facility where women attend antenatal [ANC] and postnatal care [PNC]. Researchers and CHVs worked together to recruit women and new mothers during their ANC and PNC clinic visits. Potential interviewees were then given convenient appointments for the interviews.

Pregnant women who were at least 28 weeks into their pregnancy, as confirmed by a health-care worker, were eligible for inclusion. Both first-time and experienced mothers were considered. Postpartum women between 4 to 8 weeks after childbirth were also eligible, though all interviewed mothers had babies between 4 to 5 weeks old. Mothers-in-law residing in the same household as the pregnant or postpartum women and CHVs with more than two years of community experience were included in the study. Including mothers-in-law living in the same household as pregnant or postpartum women and CHVs with over two years of community experience was intentional. In Kilifi, these individuals are often closely involved with this community's pregnant women and new mothers. Their proximity and roles allow them to provide detailed, lived experiences regarding the effectiveness and impact of weather warning systems. Potential interviewees were checked by CHVs in charge of recruitment to ensure they met the inclusion criteria, agreed to provide informed consent, and could participate in the study. To avoid bias, the researchers conducted an additional check to confirm participants' eligibility before proceeding with the actual interview.

Across Kaloleni and Rabai, we conducted 10 IDIs with pregnant women and 12 with postpartum mothers. Additionally, we held six FGDs, each with no more than 7 participants, including three groups of mothers-in-law and three groups of CHVs. Participants were provided with transportation costs and refreshments. In-depth interviews with pregnant women and new mothers were conducted in a convenient room at the local health facility. However, due to inadequate indoor space in the health facility, we held FGDs for CHVs and mothers-in-law under a tree. A debriefing statement was provided to the participants at the end of each interview.

## Ethics clearance and consenting process

This study received Ethics approval from the Aga Khan University Ethics Committee ref 2020/IERC-94 [[v2]] and National Commission for Science and Technology and Innovation Ref BAHAMAS ABS/P/20/7568, London School of Hygiene and Tropical Medicine Research Ethics Committee ref: 22685 and from Kilifi County Office ref DOM/KLF/RESCH/vol.1/66.

We employed a comprehensive approach to ensure informed consent from all participants, recognizing the diversity in literacy levels. We obtained written and verbal consent to accommodate these differences and enhance the clarity and comprehension of the study's objectives and procedures. For literate participants, informed consent was achieved through a two-step process. Initially, participants independently read detailed information sheets. Following this, researchers engaged with them to address any questions, ensuring thorough understanding. Once satisfied, participants signed and dated the consent forms, affirming their understanding of issues of potential risks and voluntary agreement to participate.

We tailored a verbal process for illiterate participants to ensure they received and understood the same information as their literate counterparts. Information sheets and consent forms were translated into the local language familiar to the participants and were read to them. These documents were then read aloud to them by a researcher in the presence of a neutral witness, either a spouse or a CHV, ensuring transparency. Each illiterate participant gave their thumbprint on the consent form. This action was witnessed and co-signed by a literate witness to authenticate further and affirm the voluntary nature of the consent. All consent forms, whether signed or thumb-printed, were collected and stored in accordance with ethical guidelines to maintain confidentiality and ensure data integrity at the Aga Khan University.

## Interviews

Interviewing commenced after piloting our study guide in an FGD with CHVs and IDI with pregnant and postpartum women. Trained social scientist researchers [AL and PK]], familiar with the study protocol and guide, conducted the interviews. All interviews were conducted at the health facilities and pregnant and postpartum women were interviewed during antenatal and postnatal visits. Researchers explained information to study participants in Kiswahili before seeking individual permission to audio-record. Researchers used an open-ended, thematically structured topic guide to prompt free-flowing conversations. Interviews lasted no more than two hours. Explored themes included community weather information seeking, reliability of sources, influences on daily behaviours, and measures taken during extreme weather events for pregnant women. The study occurred during the COVID-19 pandemic, and researchers provided participants with information on preventive measures and supplied hand sanitiser and face masks.

## Data management and analysis

Data were transcribed verbatim with the assistance of a professional transcriber who worked closely with the study researchers to check the accuracy of the transcribed data against the audio recordings. Braun and Clarke's thematic analysis framework was employed [39]. Both deductive and inductive analysis were used to inform a deeper understanding of the participants' experiences. The study team read all the field notes and a section of the transcripts to inform the codebook development. AL, PK and JN examined data on weather forecasting and resilience to heat by applying segments of the texts across all FGDs and IDIs, comparing coding outputs and charting manually in a matrix to identify sources of weather information, barriers and opportunities as illustrated in Table 1 below.

## Findings

### Characteristics of participants

We conducted IDIs with diverse participants, including 10 pregnant and 12 postpartum women, 19 mothers-in-law, and 22 CHVs. Mothers-in-law, aged between 35 and 63, had limited formal education, spoke no English, and primarily engaged in subsistence farming. Community health volunteers interviewed for this study had more than two years of experience working with the research population and worked an average of 15 hours per week. All our participants were native community members, spoke the local language, and exhibited a thorough understanding of the local culture. A summary of the participants' socio-demographic characteristics is provided in Tables 1 and 2, while a summary of themes related to weather information is presented in Table 3.

**Table 1. Socio-demographics for FGD participants.**

| Variables | Definitions | Mothers in-law | Community health volunteers |
|---|---|---|---|
| Age (years) | 16 to 25 | 0 | 0 |
| | 26 to 35 | (5.3%) | 3 (13.6%) |
| | >35 | 18 (94.7%) | 19 (86.4%) |
| Employment | Formal | 0 | 2 (9.1%) |
| | Casual | 5 (26.3%) | 11 (50%) |
| | Unemployed | 13 (68.4%) | 9 (40.9%) |
| | Retired | 1 (5.3%) | 0 |
| Level of education | High school + | 1 (5.3%) | 22 (100%) |
| | Primary school | 3 (15.8%) | 0 |
| | Uneducated | 15 (78.9%) | 0 |
| Marital status | Married | 16 (84.2%) | 18 (81.8%) |
| | Single | 0 | 1 (4.5%) |
| | Widowed | 2 (10.5%) | 3 (25%) |
| | No answer | 1 (5.3%) | 0 |
| Co-wives | Yes | 0 | 12 (54.6%) |
| | No | N/A | 7 (31.8%) |
| | No answer | 0 | 3 (13.6%) |
| Type of dwelling | Semi- permanent | 19 (100%) | 17 (77.3%) |
| | Permanent | 0 | 5 (22.7%) |
| | No answer | 0 | 0 |
| No. household members | 1 to 3 | 3 (15.8%) | 0 |
| | 4 to 6 | 6 (31.6%) | N/A |
| | >7 | 10 (52.6%) | 0 |
| No. of community health volunteer hours | 0 to 10 hours | | 14 (64%) |
| | 11 to 20 hours | | 6 (27%) |
| | 21 to 30 hours | | 1 (4.5%) |
| | No answer | | 1(4.5%) |

## I. Sources of information about heat warnings and forecasts

Our study revealed that weather warning sources encompass both modern and traditional indigenous knowledge. Participants highlighted that the local Kilifi Meteorological Centre disseminates weather warning information on heat and other severe weather through television and radio, but radio was the most commonly and widely used medium for transmitting heat alert messages.

> "...Us CHVs, we search for weather forecasting information through the radios to get news, and the radio is the most common."

While radio emerged as a widely utilized medium for disseminating weather warning messages in rural communities, some pregnant and postpartum women and mothers-in-law expressed not owning a television or radio. Consequently, they did not regularly receive weather warning messages. In community units where no households possessed a television or radio, pregnant women depended on word of mouth from peers within their communities.

> "I get weather information from people talking about it on the road or when I visit my neighbours in their houses because I have no radio or television.

**Table 2. Socio-demographics for pregnant and postpartum women.**

| Variables | Definitions | Pregnant women | Postpartum women |
|---|---|---|---|
| Age (years) | 16 to 25 | 7 (70%) | 7 (58.3%) |
| | 26 to 35 | 3 (30%) | 3 (25%) |
| | >35 | 0 | 2 (16.7%) |
| Employment | Formal | 1 (10%) | 3 (25%) |
| | Casual | 0 | 2 (16.7%) |
| | Unemployed | 9 (90%) | 7 (58.3%) |
| Level of education | High school + | 3 (30%) | 7 (58.3%) |
| | Primary school | 6 (60%) | 3 (25%) |
| | Uneducated | 1 (10%) | 2 (16.7%) |
| Marital status | Married | 9 (90%) | 9 (75%) |
| | Single | 1 (10%) | 3 (25%) |
| Co-wives | Yes | 9 (90%) | 9 (75%) |
| | No | 0 | 1 (8.3%) |
| | No answer | 1 (10%) | 2 (16.7%) |
| Type of dwelling | Semi- permanent | 9 (90%) | 6 (50%) |
| | Permanent | 1 (10%) | 5 (41.7%) |
| | No answer | 0 | 1 (8.3%) |
| No. household members | 1 to 3 | 6 (60%) | 4 (33.3%) |
| | 4 to 6 | 3 (30%) | 5 (41.7%) |
| | >7 | 1 (10%) | 3 (25%) |
| Previous pregnancies | 0 to 3 | 9 (90%) | 10 (83.3%) |
| | >4 | 9 (10%) | 2 (16.7%) |
| No. of Children | 0 to 2 | 9 (90%) | 7 (58.3%) |
| | >3 | 1 (10%) | 5 (41.7%) |

*** We excluded data for one pregnant woman due to poor audio recordings.

While smartphones serve as sources of information in certain regions of the country, our participants did not cite this channel of communication. Researchers observed that a few participants possessed mobile phones, and these devices were not smartphones. Acquiring and using smartphones comes at a cost and searching for messages on these phones would necessitate proficiency in English, a language many of our interviewees did not speak.

**Table 3. Summary of themes on weather information.**

| Sources of weather information | Meteorological forecasts (Radio and Television). | Indigenous knowledge of weather. |
|---|---|---|
| | | Rainmakers [individuals who traditionally control the rain]. |
| | Oral [word of mouth through chiefs, CHVs, government officials, friends and neighbours]. | Behaviour of insects (such as butterflies and ants). |
| | | Behaviour of frogs; observing clouds and changes in specific trees, e.g. Baobab |
| Barriers | Limited ownership of TV and radio; normalization of heat;inaccurate or inconsistent forecasts; lack of trust in meteorological forecast; reluctance of younger people to learn indigenous knowledge from older people; lack of awareness of the availability of official forecasts, and education and language barriers [Weather information is communicated in English]. | |
| Benefits of weather forecasts | Helps mothers to Plan in advance; empowers CHVs to disseminate messages to the community on heat risks and possible measures and allows local chiefs to disseminate the messages to the communities on time. | |

*Indigenous knowledge.* Participants noted that indigenous methods, such as observing clouds, consulting rainmakers, and studying the behavioural patterns of amphibians [frogs and toads], birds, cows, and certain plants, are still used for early warning to predict rainfall patterns, drought and imminent cyclones. Postpartum women described the presence of rainmakers within these communities who predict different seasons.

> *"There are rain makers who predict the weather, who will predict that this month it will be cooler, more rain, or sunny season,"* and the potential of having rain is shown by the presence of many butterflies, *"...when there are many butterflies and ants, they [the community] will know that it will rain".*

Mothers-in-law shared their knowledge of weather forecasting, explaining how observing clouds, changes in specific trees, and the behaviours of frogs helped them understand the transition of weather patterns throughout the seasons. For instance, seeing red clouds, the filling of leaves on a red baobab tree, or the croaking of frogs served as signs of impending heavy rainfall.

> *"... and to know if it is about to rain, we observe heavy clouds in the morning; when we see red clouds on this part [pointing East], and the sun shall have gone to the other part [pointing West] ... you will then see the red clouds, and that is already a sign of rain, with heavy clouds...".*

> *"...there is a particular tree found in the grass that when you see it growing, or a dry baobab tree, and when you see them filling their leaves, that is a clear indication that it is about to rain. That is according to our culture. ... and then when you hear frogs, there is no water, but you will hear them croak, it is a clear sign that the rains are about to fall, and it usually doesn't exceed two weeks before it starts to rain..."*

The indigenous methods are community-specific and inform the communities of seasonal rains so that they can prepare for different activities performed during different seasons, such as planting. They did not address temperature extremes, but only rainfall. The crucial role played by local chiefs and other key officials from various government ministries in disseminating rainfall forecasts by word of mouth was emphasised by CHVs.

> *"...we get information about early weather warnings from public barazas [meetings]. In these public meetings, you will find people from the Ministry of Agriculture there, who tell us to be aware that in such and such a month, there shall be rains so that we prepare to plant crops in advance..."*

## II. Low perception of heat extremes and forecast information

Some responses from mothers-in-law suggest that heat warnings may not be useful. This may be because the area experiences consistently high temperatures [above 23°C] throughout the year, which many community members have become accustomed to.

> *"It is always hot; even when it rains, it is hot; at night, it is hot; even when they say it will rain, it is still hot, does this information change anything?"*

Researchers observed a low perception of heat-related health risks, a concern acknowledged and recognized by community health workers who reported instances of "*mothers walking*

*with one-day-old babies in the heat."* The diminished awareness of heat risks may stem from a lack of information, knowledge, and awareness regarding the existence of weather warning messages and how heat affects health. In response to inquiries about their frequency of searching for weather information, some postpartum women interviewed for the study provided varied responses, ranging from *"I have never heard about early weather warnings"* to *"I don't know"* or *"I don't seek this information."*

Some participants had concerns about the accuracy of alerts, which would explain why indigenous systems continue to be used alongside the forecasts from the Met Service.

> *"... they [the Department of Meteorology] will inform us that there will be moisture in the morning and that it will remain dry in the afternoon in the coastal areas. When we listened to them [meteorologists] at the beginning of the year, what they predicted happened exactly the same. However, we were surprised recently as their weather information was wrong, as it rained so heavily. Everyone was surprised in the month of February..."*
>
> *Focus group discussions, Mothers-in-law, Makombaini*

Our area of study was rural, with low literacy levels and low perceptions that arise from the lack of understanding around the confidence they placed in weather forecasts. The community members expect the weather forecast on rainfall to be accurate, which was not always the case. Similarly, communities may tend to trust temperature forecasts as it is always hot throughout the year. Moreover, communities may also lack understanding around the fact that forecasts are more likely to be accurate at short lead times [i.e. forecasts for the same day or the following day] compared with forecasts for the months ahead].

## III. Responding to weather warnings

Our research team investigated participants' perspectives on the perceived utility of receiving weather forecasts and warnings concerning impending extreme heat or heavy rainfall. CHVs highlighted that, given their involvement with communities, such warnings assist them in pre-planning their activities.

> *"...so during the rainy season, you can plan that today you will do this work, and then tomorrow you will do that other work. It [early weather information] helps you plan your activities between your work and the CHV work."*

Further, the CHVs noted that based on their experience, whenever they receive information regarding imminent extreme weather, they use this knowledge to advise mothers and the community at large on taking protective measures; examples of such actions include *"storing more food"*, *staying inside the house when it is hot"*, *"drinking water"*, and *"avoiding walking in the sun*. CHV uses weather information to guide mothers in sleeping under mosquito nets with their babies and conserving wastewater in kitchen gardens during dry seasons. The CHVs reported sharing information with pregnant women's support networks, such as mothers-in-law, to assist their daughters-in-law during high temperatures.

> *"...when I get that information about hot temperatures, I advise pregnant women, postnatal mothers, and even all the people that since they will not be using clothes at night, they should sleep under the mosquito nets. Because during hot periods, mosquitoes are many. So that is what I normally advise them."*

*"We normally help them [mothers], . . . for the pregnant woman or a breastfeeding mother that has a kitchen garden, we tell them that since we are going to have a dry season, and for them to get some green vegetables "[mchicha]" close to the house, they should conserve the domestic wastewater to use to irrigate the crops in the garden so that during the dry spell, she can get the vegetables "[mchicha]" that would be beneficial to her."*

Pregnant and postpartum women emphasized the crucial need for accurate and timely early weather information. They expressed a strong desire to receive such information, highlighting its importance in preparing for various weather patterns. For instance, information about an impending drought received through the radio was cited as particularly beneficial for organizing and planning. Being informed about the upcoming drought enables them to proactively secure water and food resources. Additionally, they mentioned the strategic planting of more food during the current rainy season to ensure a sufficient supply during the anticipated drought.

*"It can help in organising yourself. For example, if I get information from the radio about a drought that will come for two months consecutively, I will organise myself . . . I will know where to get water and food. Like now it is raining, I will have to plant more food so that when the drought season comes, I will have enough food to help me."*

*IDI, postpartum*

Having information on the weather would allow pregnant mothers to schedule their duties more effectively. Specifically, during cooler times like morning and evening, they can complete their chores efficiently without being adversely affected by the heat. This planning allows for a more relaxed time during the hotter parts of the day.

*"You will do your duties early, which you cannot do when scorching because it will affect you. You will hurry in doing them [chores] in the morning and evening so that you can relax during the day." IDI, pregnant*

## Discussions

Our research identified gaps in the delivery of early warning and weather information. The information channels are restricted to television and radio, implying that many rural women without access to these communication methods do not receive notification. Moreover, when information is delivered in English or through online platforms, this has a limited reach because many rural mothers are unlikely to own modern phones, need help reading, and may need help understanding English.

Some mothers interviewed for this study reported having no idea what early warnings were or having not searched for weather information. The unequal gender access to climate services and information in sub-Saharan Africa has been previously documented in the literature [40–42]. Studies show that men are likely to listen to the radio compared to women, more likely to own smartphones and access weather information [41]. Women's daily household chores deprive them of time to listen to the radio [43]. The importance of transmitting weather forecasts and other climate services many times a day has been highlighted [42] to provide opportunities for mothers busy with household chores to access this information. Our results show the need to educate the communities to seek weather information and translate into local languages such as Giriama and Rabai.

Our findings illustrate the need for the National Governmental structures and the Department of Meteorology to regularly leverage the existing platform of CHVs, religious gatherings, engaging health facility workers and local chiefs ['Barazas'], like weekly community meetings. This would ensure these messages reach the most vulnerable community members. Although sharing information by word-of-mouth can be distorted, in Kilifi, it will affect behaviour, especially for those without televisions or radios. The successful use of community key influencers and social media to disseminate early warnings and heat risk information is discussed in the literature. [44, 45] with conclusions based on observations in this study. Researchers for this study observed that in Kilifi, mothers have limited time to attend events where weather information is disseminated, such as local public chiefs meetings, due to childcare responsibilities. In such cases, access to information can be enhanced if services are provided in their communities where childcare and other household chores are close [43, 46, 47].

Despite the barriers faced in accessing this information, participants in our study noted the importance of receiving timely messages to assist them in taking protective measures and preparing for impending extreme weather conditions. For instance, heat and weather warnings will help pregnant and postpartum mothers to attend antenatal and postnatal activities early in the morning; task shifting to perform outdoor chores such as fetching water, farming, and searching for firewood early in the morning; shopping, wearing light clothes and avoiding walking with newborns in the heat.

Participants emphasized the ongoing reliance on indigenous knowledge for weather forecasts. However, the accuracy of these forecasts is likely to change as the weather deviates from historical norms, becoming more extreme and variable. These findings align with research conducted in other parts of sub-Saharan Africa [48–50]. It is noteworthy that these methods are predominantly retained and reinforced by the older generation. The study researchers observed a need for greater transmission of this knowledge to younger generations. This is particularly crucial in communities facing economic deprivation where many residents cannot afford a radio, as these indigenous methods may persist for an extended period. Local policy makers in Kilifi have emphasized the importance of preserving indigenous knowledge and the need to put measures in place to preserve these traditions by providing training to the younger generations [51]. Similar concerns have been raised in research in Zimbabwe [52] and Ethiopia [53]. Indigenous weather predictors could collaborate with the local chiefs and disseminate these messages on market days or during the chief's weekly public address, alongside weather warning information from the Kilifi Meteorological Centre.

Although our participants talked about channels of weather warnings, it wasn't clear whether heat health warning systems were available. We recommend that these messages be codesigned with community members, including indigenous local predictors, women groups, and local and civil society organisations. It should account for pregnant and postpartum women's needs. More research is needed to find out if traditional weather signs, like frogs croaking before rain, still work in a changing climate. It is also possible that as weather patterns change, these signs might change too.

Our findings have limitations and cannot be generalized to other areas, but they can serve as valuable insights to guide future research. Researchers encountered various challenges, including the reluctance of mothers to discuss their experiences with weather information and early warnings. Many young mothers were unaware of such warnings, and several older women interviewed for the study were more at ease discussing indigenous methods of weather information.

## Conclusions

There is an urgent need for improved communication and delivery of weather information and warnings to guide community-level mitigation measures. We found serious gaps in information regarding heat forecasts. Broadcasting through television and radio is currently the common method to deliver such messages, however messages to the community´s should be segmented and translated into local languages. Community health volunteer´s network with key influencers such as the local government administrative chiefs, religious leaders, opinion leaders, and facility healthcare workers should be engaged to disseminate weather warning messages.

## Acknowledgments

We express our gratitude to all the research participants in Kaloleni and Rabai. Special thanks to Felix Agoi, Sophie Chabeda and the CHVs for their valuable assistance in recruiting study participants. Our appreciation extends to all the members of the CHAMNHA consortium.

## Author Contributions

**Conceptualization:** Adelaide Lusambili, Veronique Filippi, Britt Nakstad, Nathalie Roos, Peter Khaemba, Sari Kovats.

**Data curation:** Adelaide Lusambili, Julian Natukunda, Peter Khaemba.

**Formal analysis:** Adelaide Lusambili, Veronique Filippi, Julian Natukunda, Peter Khaemba.

**Funding acquisition:** Britt Nakstad, Sari Kovats.

**Investigation:** Adelaide Lusambili, Peter Khaemba, Sari Kovats.

**Methodology:** Adelaide Lusambili, Peter Khaemba.

**Project administration:** Adelaide Lusambili.

**Resources:** Cathryn E. Birch, John H. Marsham.

**Supervision:** Sari Kovats.

**Validation:** Adelaide Lusambili, Veronique Filippi, Britt Nakstad, Julian Natukunda, Cathryn E. Birch, John H. Marsham, Nathalie Roos, Peter Khaemba, Sari Kovats.

**Visualization:** Adelaide Lusambili, Veronique Filippi, Britt Nakstad, Julian Natukunda, Cathryn E. Birch, John H. Marsham, Nathalie Roos.

**Writing – original draft:** Adelaide Lusambili, Nathalie Roos.

**Writing – review & editing:** Adelaide Lusambili, Veronique Filippi, Britt Nakstad, Julian Natukunda, Cathryn E. Birch, John H. Marsham, Nathalie Roos, Sari Kovats.

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
