## [Decision Letter · Decision Letter 0]

25 Jun 2024

PONE-D-24-16377A Qualitative Study of Heat Early Weather Warnings for Pregnant Women and Children in Kilifi, KenyaPLOS ONE

Dear Dr. Lusambili,

Thank you for submitting your manuscript to PLOS ONE. After careful consideration, we feel that it has merit but does not fully meet PLOS ONE’s publication criteria as it currently stands. Therefore, we invite you to submit a revised version of the manuscript that addresses the points raised during the review process.

Please ensure you read all comments from all reviewers carefully and respond in kind to their comments and suggestions. Also please ensure particular focus is given to the clarity and overall direction of the manuscript as well as description and justification of the methods, as described by Reviewers 2 and 3.

We look forward to receiving your revised manuscript.

Kind regards,

James Colborn

Academic Editor

PLOS ONE

Journal Requirements:

2. In the ethics statement in the Methods, you have specified that verbal consent was obtained. Please provide additional details regarding how this consent was documented and witnessed, and state whether this was approved by the IRB

4. Thank you for stating the following financial disclosure: "This work was supported by the Natural Environment Research Council (NERC) [grant numbers NE/T013613/1, NE/T01363X/1]; Research Council of Norway (RCN) [grant number 312601]; The Swedish Research Council for Health, Working Life and Welfare in collaboration with the Swedish Research Council (Forte) [grant number 2019-01570]; and the National Science Foundation (NSF) [grant number ICER-2028598]; coordinated through a Belmont Forum partnership."

5. Please note that in order to use the direct billing option the corresponding author must be affiliated with the chosen institute. Please either amend your manuscript to change the affiliation or corresponding author, or email us at plosone@plos.org with a request to remove this option.

Reviewers' comments:

Reviewer's Responses to Questions

**Comments to the Author**

1. Is the manuscript technically sound, and do the data support the conclusions?

Reviewer #1: Yes

Reviewer #2: Yes

Reviewer #3: Partly

2. Has the statistical analysis been performed appropriately and rigorously? 

Reviewer #1: N/A

Reviewer #2: Yes

Reviewer #3: N/A

3. Have the authors made all data underlying the findings in their manuscript fully available?

Reviewer #1: No

Reviewer #2: No

Reviewer #3: Yes

4. Is the manuscript presented in an intelligible fashion and written in standard English?

Reviewer #1: Yes

Reviewer #2: Yes

Reviewer #3: Yes

5. Review Comments to the Author

Reviewer #1: Review comments

General comment

I commend the Authors for conducting the study and the well written manuscript. The study and its findings are of relevance to the community and its inhabitants. Bearing in mind the number of interviews and FGDs that were conducted, perhaps I expected mores quotes in the manuscript.

The dates Internet articles were assessed were not included in the reference section. Also, PLoS ONE makes use of square brackets for reference numbers.

The Authors should also cross-check the length and sub-headings of the abstract section to ensure that it conforms to the guidelines of the Journal.

I recommend that the manuscript be accepted for publication.

Reviewer #2: A strong paper with just a few edits suggested in the attached document.

Reviewer #3: First congratulations to the authors for the well- thought-out paper. The manuscript highlights a critical issue given the increasing recognition of the linkages between climate and health. That said, below are some suggestions to further strengthen the manuscript.

Overall comment

A few reflection issues for the authors:

i) The title of this study is "A Qualitative Study of Heat Early Weather Warnings for Pregnant Women and Children in Kilifi, Kenya". This implies that your primary target population is pregnant women and children. I am struggling with the value that CHVs and Mothers in law bring to your manuscript, since they are not your primary study population, yet their perspectives are given prominence. For instance, the authors first finding on sources of information highlights how CHVs access information. The section on indigenous information highlights how mothers in law access information. Unless these other players (CHVs etc.) are sharing experiences about how pregnant and postpartum women access information, it is not very helpful for this manuscript if they are describing their own experiences. I would suggest the authors prioritize the experiences of pregnant and post-partum women and exclude these other populations. The alternative would be to revise your study title, background and research questions - so that you expand beyond just pregnant and post-partum women, and also include the mothers in law, CHVs, spouses etc.

ii) I also didn't see the value of interviewing male spouses, yet they don't appear anywhere in the findings. They should either be included in the findings or excluded altogether if they don't fit this manuscript.

iii) The title prioritizes pregnant women and children. However, not much in the findings covers children. Also, the findings cover a lot about post-partum women, yet they are missing in the title - Reading the findings, a befitting population for the title would be pregnant and post-partum women, rather than pregnant women and children.

Specific Comments

1. Abstract - The abstract is well written.

2. Introduction: The introduction provides a good summary of the literature, clearly pinpointing the problem addressed through the study.

3. Methods: There are opportunities for the authors to strengthen the methodology so that the study is easily reproducible.

i) The authors introduce the methods used for data collection in their study design, but don't mention their sample size. It would have been helpful to include the number of FGDs and IDI's they conducted in the section where they describe the study design.

ii) The participant recruitment can also be improved. The statement by the authors "Local health facility healthcare workers and certified Community Health Volunteers (CHVs) identified potential interviewees in the communities according to our inclusion and exclusion criteria" is insufficient. What was this inclusion criteria? How big was the catchment area - The two sub-Counties mentioned under study site are vast - Did the study cover the whole sub-County? How many health workers and CHVs did they use for recruitment? How did they narrow down to the CHV or health worker to work with? How many pregnant and postpartum women were within the catchment area? What criteria or procedures did they follow to arrive at the study participant? What sampling approach did the authors use? If the authors can answer these questions, it may help improve their methodology.

iii) The section on interviewing states "pregnant and post-partum women were interviewed during antenatal and

postnatal visits at health facilities as part of their routine appointments". Based on this statement, were they recruited at the household then interviewed at the facility or were they recruited in the facilities? A clarification on this would be helpful for the reader.

4. Findings: I would suggest that the authors prioritize the findings that address the problem highlighted in the background - How heat affects maternal and child health outcomes. Some of the findings, although useful, steer away from this focus and address effects of heat and early warning systems in general.

i) It would be helpful if the authors could provide a table summarizing the participants socio-demographics. This would assist the reader make comparisons across the different groups beyond the summary they have provided.

ii) The authors could provide de-identified summary socio-demographic information to accompany each verbatim quote e.g. (28-year-old CHV, Kaloleni).

iii) Some of the findings such as how CHV's use weather warnings to plan their work doesn't appear a very good fit. Neither is the finding around drought the most ideal given the focus of the manuscript on maternal health outcomes.

5. Discussion: The authors should ensure the discussion is anchored on the findings and avoid introducing issues in the discussion that were not included in the findings. For instance, the gender differences in access to information is first included in the discussion. Same to the issue of women's participation in public meetings.

6. Conclusion: It is also important for the authors to draw conclusions that are relevant to the study population and beyond. For instance, the findings highlighted gaps in access to information through radio and television since some participants did not own these, and low mobile phone ownership, yet conclusions affirm these as the methods to be adopted. I would suggest that the authors relook at their findings when drawing conclusions and stratify their conclusions to the different subgroups.

6. PLOS authors have the option to publish the peer review history of their article (what does this mean?). If published, this will include your full peer review and any attached files.

Reviewer #1: **Yes: **EDMUND NDUDI OSSAI

Reviewer #2: **Yes: **Beth A Tippett Barr

Reviewer #3: No

---

## [Author Response · Author response to Decision Letter 0]

23 Sep 2024

RESPONSE TO REVIEWERS

Thank you. We have checked the manuscript and addressed all this, throughout the manuscript.

2. In the ethics statement in the Methods, you have specified that verbal consent was obtained. Please provide additional details regarding how this consent was documented and witnessed, and state whether this was approved by the IRB

We have provided detailed information on how we conducted verbal ethics. Line 675 to 691 in the tracked changes draft.

This has been addressed.

4. Thank you for stating the following financial disclosure: "This work was supported by the Natural Environment Research Council (NERC) [grant numbers NE/T013613/1, NE/T01363X/1]; Research Council of Norway (RCN) [grant number 312601]; The Swedish Research Council for Health, Working Life and Welfare in collaboration with the Swedish Research Council (Forte) [grant number 2019-01570]; and the National Science Foundation (NSF) [grant number ICER-2028598]; coordinated through a Belmont Forum partnership."

Please state what role the funders took in the study. If the funders had no role, please state: 

If this statement is incorrect, you must amend it as needed. 

Thank you we have added this statement in the cover letter.

5. Please note that in order to use the direct billing option the corresponding author must be affiliated with the chosen institute. Please either amend your manuscript to change the affiliation or corresponding author, or email us at plosone@plos.org with a request to remove this option. 

The corresponding author is

Dr Adelaide Lusambili

Email: adelaidelusambili@gmail.com

Thank you. This has been addressed.

We have included Tables 1 and 2 at the end of the manuscript. These provide socio-demographic information for pregnant and postpartum women, CHVs, and mothers-in-law.

8. Please review your reference list to ensure that it is complete and correct. If you have cited papers that have been retracted, please include the rationale for doing so in the manuscript text, or remove these references and replace them with relevant current references. Any changes to the reference list should be mentioned in the rebuttal letter that accompanies your revised manuscript. If you need to cite a retracted article, indicate the article’s retracted status in the References list and also include a citation and full reference for the retraction notice. We have updated references by including dates that we retrieved the documents. 

Reviewers' comments:

Reviewer's Responses to Questions

Comments to the Author

9. Is the manuscript technically sound, and do the data support the conclusions?

Reviewer #1: Yes

Reviewer #2: Yes

Reviewer #3: Partly

10. Has the statistical analysis been performed appropriately and rigorously?

Reviewer #1: N/A

Reviewer #2: Yes

Reviewer #3: N/A

11. Have the authors made all data underlying the findings in their manuscript fully available?

Reviewer #1: No

Reviewer #2: No

Reviewer #3: Yes

12. Is the manuscript presented in an intelligible fashion and written in standard English?

Reviewer #1: Yes

Reviewer #2: Yes

Reviewer #3: Yes

13. Review Comments to the Author

Reviewer #1: Review comments

General comment

14. I commend the Authors for conducting the study and the well written manuscript. The study and its findings are of relevance to the community and its inhabitants. Bearing in mind the number of interviews and FGDs that were conducted, perhaps I expected mores quotes in the manuscript.

The dates Internet articles were assessed were not included in the reference section. Also, PLoS ONE makes use of square brackets for reference numbers.

The Authors should also cross-check the length and sub-headings of the abstract section to ensure that it conforms to the guidelines of the Journal.

• We have included the dates internet articles were accessed.

• We have put square brackets for reference numbers

• We have amended the length and sub headings of the abstract section to conform with the guidelines of the journal.

I recommend that the manuscript be accepted for publication. Thank you

15. Reviewer #2: A strong paper with just a few edits suggested in the attached document.

Abstract – overall this is well-written, but please ensure it’s within the journal’s word limit. 

Thank you . This has been done.

Introduction 

• Please include what the UN early warning for all is – what’s their aim more specifically?

We have now included more details about the Early Warning for All Initiative

Methods – 

• Subheading “Interviewing” should be “Interviews”. We have addressed this.

• Remove period after ‘criteria’ subheading. Thank you. This has been addressed.

Results – 

• Remove “scorching” from your narrative in the sentence before it also appears in the IDI quote

Discussion 

We have now done this. 

• Check consistency of terminology compared to earlier in the paper “early warning weather” vs “early weather warning: ” @team@still pending

Thank you. We have now clarified this in the text. This papers addressed both weather information (routine forecasts) and weather alerts or early warnings. 

• It seems worth stating that more research is needed to ascertain whether traditional weather indications stay true in the changing climate (ie. Frogs croaking <2 weeks before rains). Even when weather patterns shift, it’s likely (?). 

This has been added in the discussions section

Reviewer #3: First congratulations to the authors for the well- thought-out paper. The manuscript highlights a critical issue given the increasing recognition of the linkages between climate and health. That said, below are some suggestions to further strengthen the manuscript.

Overall comment

A few reflection issues for the authors:

• The title of this study is "A Qualitative Study of Heat Early Weather Warnings for Pregnant Women and Children in Kilifi, Kenya". This implies that your primary target population is pregnant women and children. I am struggling with the value that CHVs and Mothers in law bring to your manuscript, since they are not your primary study population, yet their perspectives are given prominence. For instance, the authors first finding on sources of information highlights how CHVs access information. The section on indigenous information highlights how mothers in law access information. Unless these other players (CHVs etc.) are sharing experiences about how pregnant and postpartum women access information, it is not very helpful for this manuscript if they are describing their own experiences. I would suggest the authors prioritize the experiences of pregnant and post-partum women and exclude these other populations. The alternative would be to revise your study title, background and research questions - so that you expand beyond just pregnant and post-partum women, and also include the mothers in law, CHVs, spouses etc. I also didn't see the value of interviewing male spouses, yet they don't appear anywhere in the findings. They should either be included in the findings or excluded altogether if they don't fit this manuscript.

• We have excluded male spouses as they did not contribute to this theme. The title has been revised to include postpartum women.

• iii) The title prioritizes pregnant women and children. However, not much in the findings covers children. Also, the findings cover a lot about post-partum women, yet they are missing in the title - Reading the findings, a befitting population for the title would be pregnant and post-partum women, rather than pregnant women and children.

Thank you. You made good observations on the efficacy of the current title. Our aim was to gather community perspectives on weather warning systems during pregnancy and postpartum. We did so by gathering different views from those who work closely with pregnant and postpartum women, such as CHVs and mothers-in-law. We feel it important that care givers, and those who advise pregnant and post-partum women, also have access to weather information and heat warnings

Specific Comments

1. Abstract - The abstract is well written. Thank you.

2. Introduction: The introduction provides a good summary of the literature, clearly pinpointing the problem addressed through the study. Thank you

3. Methods: There are opportunities for the authors to strengthen the methodology so that the study is easily reproducible. 

The methodology has been strengthened. See line 652 to line 684.

i) The authors introduce the methods used for data collection in their study design, but don't mention their sample size. It would have been helpful to include the number of FGDs and IDI's they conducted in the section where they describe the study design. We have now included the sample size line 652 and 684. Further the sample size table is below the references.

ii) The participant recruitment can also be improved. The statement by the authors "Local health facility healthcare workers and certified Community Health Volunteers (CHVs) identified potential interviewees in the communities according to our inclusion and exclusion criteria" is insufficient. What was this inclusion criteria? How big was the catchment area - The two sub-Counties mentioned under study site are vast - Did the study cover the whole sub-County? How many health workers and CHVs did they use for recruitment? How did they narrow down to the CHV or health worker to work with? How many pregnant and postpartum women were within the catchment area? What criteria or procedures did they follow to arrive at the study participant? What sampling approach did the authors use? If the authors can answer these questions, it may help improve their methodology.

Thank you. We have noted your insightful observations, further improved the methods section in the document, and addressed some of the issues you raised that were relevant to the current study.

iii) The section on interviewing states "pregnant and post-partum women were interviewed during antenatal and postnatal visits at health facilities as part of their routine appointments". Based on this statement, were they recruited at the household then interviewed at the facility or were they recruited in the facilities? A clarification on this would be helpful for the reader. 

We have made this clear by stating that we recruited them at the health facility during antenatal care and post natal care visits and set a convenient day and time for their appointment thereafter.

4. Findings: I would suggest that the authors prioritize the findings that address the problem highlighted in the background - How heat affects maternal and child health outcomes. Some of the findings, although useful, steer away from this focus and address effects of heat and early warning systems in general.

Thank you. The purpose of this paper is to report () - Community members' from Kilifi early weather information-seeking behaviour, identify available sources, assess their reliability and utility, and examine both their support measures and its influence on behaviour. Maternal and child outcomes are reported in our many other studies. Follow this link: https://www.lshtm.ac.uk/research/centres-projects-groups/chamnha#publications

i) It would be helpful if the authors could provide a table summarizing the participants socio-

demographics. This would assist the reader make comparisons across the different groups beyond the summary they have provided.

We have added a supplementary table of socio-demographics as requested.

iii) Some of the findings such as how CHV's use weather warnings to plan their work doesn't appear a very good fit. Neither is the finding around drought the most ideal given the focus of the manuscript on maternal health outcomes. 

This finding is particularly relevant to caregiving in the community, as CHVs play a crucial role in working with mothers and delivering vital messages, especially in geographically disconnected areas like Kilifi. In such regions, the role of CHVs is central to achieving positive maternal outcomes.

5. Discussion: The authors should ensure the discussion is anchored on the findings and avoid introducing issues in the discussion that were not included in the findings. For instance, the gender differences in access to information is first included in the discussion. Same to the issue of women's participation in public meetings. 

This is well noted. We would like to note that we are locating our study in the larger body of literature that addresses these barriers and possible solutions that could improve access to weather warning information.

6. Conclusion: It is also important for the authors to draw conclusions that are relevant to the study population and beyond. For instance, the findings highlighted gaps in access to information through radio and television since some participants did not own these, and low mobile phone ownership, yet conclusions affirm these as the methods to be adopted. I would suggest that the authors relook at their findings when drawing conclusions and stratify their conclusions to the different subgroups.

We have edited the conclusion. Thank you.

---

## [Decision Letter · Decision Letter 1]

31 Oct 2024

Community Perspectives of Heat and Weather Warnings for Pregnant and Postpartum Women in Kilifi, Kenya

PONE-D-24-16377R1

Dear Dr. Lusambili,

We’re pleased to inform you that your manuscript has been judged scientifically suitable for publication and will be formally accepted for publication once it meets all outstanding technical requirements.

Kind regards,

James Colborn

Academic Editor

PLOS ONE

Additional Editor Comments (optional):

Reviewers' comments:

Reviewer's Responses to Questions

**Comments to the Author**

1. If the authors have adequately addressed your comments raised in a previous round of review and you feel that this manuscript is now acceptable for publication, you may indicate that here to bypass the “Comments to the Author” section, enter your conflict of interest statement in the “Confidential to Editor” section, and submit your "Accept" recommendation.

Reviewer #1: All comments have been addressed

Reviewer #2: All comments have been addressed

2. Is the manuscript technically sound, and do the data support the conclusions?

Reviewer #1: Yes

Reviewer #2: Yes

3. Has the statistical analysis been performed appropriately and rigorously? 

Reviewer #1: N/A

Reviewer #2: Yes

4. Have the authors made all data underlying the findings in their manuscript fully available?

Reviewer #1: No

Reviewer #2: Yes

5. Is the manuscript presented in an intelligible fashion and written in standard English?

Reviewer #1: Yes

Reviewer #2: Yes

6. Review Comments to the Author

Reviewer #1: Review comments

General comments

Once again, I commend the Authors for the well written manuscript as submitted. Keep up the good work.

I recommend that the manuscript should be accepted for publication.

Observations.

1. Even though acronyms were introduced when they were used for the first time in the abstract section, the case was not the same in the body of the manuscript. Authors should refer to lines 149 and 262.

2. Tables 1 and 2 should be titled ‘Participants’ profile’ instead of ‘Socio-demographics’.

Reviewer #2: Thank you for addressing all of the comments provided in the first review.

7. PLOS authors have the option to publish the peer review history of their article (what does this mean?). If published, this will include your full peer review and any attached files.

Reviewer #1: **Yes: **EDMUND NDUDI OSSAI

Reviewer #2: No

---

## [Editor Report · Acceptance letter]

8 Nov 2024

PONE-D-24-16377R1 

PLOS ONE

Dear Dr. Lusambili, 

I'm pleased to inform you that your manuscript has been deemed suitable for publication in PLOS ONE. Congratulations! Your manuscript is now being handed over to our production team.

Kind regards, 

on behalf of

Dr. James Colborn 

Academic Editor

PLOS ONE